# Rice Mapping and Growth Monitoring Based on Time Series GF-6 Images and Red-Edge Bands

**Xueqin Jiang** [1] , **Shenghui Fang** [1,*], **Xia Huang** [1], **Yanghua Liu** [2] **and Linlin Guo** [2]

1   School of Remote Sensing and Information Engineering, Wuhan University, Wuhan 430079, China; xueqinjiang@whu.edu.cn (X.J.); huangxia@whu.edu.cn (X.H.)
2   Piesat Information Technology Co., Ltd., Beijing 100000, China; liuyanghua@piesat.cn (Y.L.); guolinlin@piesat.cn (L.G.)
*   Correspondence: shfang@whu.edu.cn

**Abstract:** Accurate rice mapping and growth monitoring are of great significance for ensuring food security and agricultural sustainable development. Remote sensing (RS), as an efficient observation technology, is expected to be useful for rice mapping and growth monitoring. Due to the fragmented distribution of paddy fields and the undulating terrain in Southern China, it is very difficult in rice mapping. Moreover, there are many crops with the same growth period as rice, resulting in low accuracy of rice mapping. We proposed a red-edge decision tree (REDT) method based on the combination of time series GF-6 images and red-edge bands to solve this problem. The red-edge integral and red-edge vegetation index integral were computed by using two red-edge bands derived from GF-6 images to construct the REDT. Meanwhile, the conventional method based on time series normalized difference vegetation index (NDVI), normalized difference water index (NDWI), enhanced vegetation index (EVI) (NNE) was employed to compare the effectiveness of rice mapping. The results indicated that the overall accuracy and Kappa coefficient of REDT ranged from 91%–94% and 0.82–0.87, improving about 7% and 0.15 compared with the NNE method. This proved that the proposed technology was able to efficiently solve the problem of rice mapping on a large scale and regions with fragmented landscapes. Additionally, two red-edge bands of GF-6 images were applied to monitor rice growth. It concluded that the two red-edge bands played different roles in rice growth monitoring. The red-edge bands of GF-6 images were superior in rice mapping and growth monitoring. Further study needs to develop more vegetation indices (VIs) related to the red-edge to make the best use of red-edge characteristics in precision agriculture.

**Keywords:** rice mapping; red-edge; growth monitoring; GF-6; time series

## 1. Introduction

Rice is one of the most important staple food in the world, especially in China. There are more than half of the population worldwide taking rice as their staple food [1,2]. Timely and accurate monitoring of rice growth is of great significance for food security and yield estimation [3,4]. Furthermore, it plays an important role in maintaining food stability and formulate relevant policies [5].

In recent decades, due to the increase of population and food demand, the rapid expansion of rice fields, especially in the south of China, led to great pressure on cultivated land, water, and environmental resources [6,7]. Remote sensing (RS) technology provides a viable approach for large area rice mapping. Different remotely sensed data (moderate-resolution imaging spectroradiometer (MODIS), Landsat, Sentinel-2, et al.) were used to monitor global and regional rice grow th [8,9]. Although there are some successful applications, rice mapping in the large area remains challenging in fragmented landscapes, such as the rice-planting areas in southern China [7,10,11]. Multiple data sources and approaches for rice mapping have been utilized and these methods can be generally divided into two categories [9]. Some mapped paddy rice using machine learning algorithm based on a

single image, including supervised classifiers like maximum likelihood [12], support vector machine (SVM) [13,14], artificial neural network (ANN) [15], and unsupervised classifiers like Iterative Self-Organizing Data Analysis Technique (ISODATA) [16]. However, different classification results would be generated with various training sets and images [2,9,17]. Besides, some phenological characteristics and time series vegetation index (VI) were applied to rice mapping [2,18–20]. There is a special physical characteristic in rice fields, which is a mixture of water, soil, and seedlings in the transplanting period. Therefore, the land surface water index (LSWI), normalized difference water index (NDWI), normalized difference vegetation index (NDVI), and enhanced vegetation index (EVI) were used to map rice [18,19,21–23]. At present, algorithms based on transplanting have been widely used in rice mapping. For example, Xiao et al. combined the rice transplanting period with phenological characteristics to map rice in South and Southeast Asia [24]. Shi J J et al. developed an algorithm for detection and estimation of the transplanting and flooding periods of paddy rice using a combination of EVI and LSWI, with the coefficient of determination ($R^2$) value of 0.847 [25]. Compared with a single time image, more information on time series images can be mined, which is helpful to obtain phenological information and reduce classification errors [2,9]. In southern China, many crops exist during the growing stage of rice, resulting in a complex planting structure. Crops with the corresponding period of rice have similar phenological dynamics and vegetation cover change patterns, which will lead to confusion between crops and rice. For various crops, trends of red-edge bands differ in the same growth period. Therefore, it can be considered to distinguish rice from crops by using the variation of red-edge bands in different growth periods. At present, it is rarely used to map rice with red-edge bands.

Worldview-2, Worldview-3, Rapideye, Sentinel-2, and GF-6 satellites all contain red-edge band sensors. Spatial resolutions of Worldview-2, Worldview-3, and Rapideye are 1.8 m, 1.24 m, and 5 m, respectively, which are suitable for small area rice mapping [26–28]. Images of Sentinel-2 MSI contain a spatial resolution of 10 m–60 m, a width of 290 km, and a revisit period of 5 days (A/B double satellite) and 10 days (single satellite), respectively. It has been widely used in regional and large area rice mapping. For example, Cai Y et al. used the random forest (RF) method based on Sentinel-2 MSI images, time-series NDVI, and phenological data to map rice with the overall accuracy and kappa coefficient were higher than 95% and 0.93, respectively [29]. Rad A M et al. presented a new automatic rule-based method combining crop phenology-time normalized vegetation index and time series of Sentinel-2 imagery with the kappa coefficient of 0.70 [30]. Moreover, MODIS and Landsat are the most widely used satellite datasets in large area rice mapping [18,22,24,31]. MODIS data has a high revisit frequency. However, its rough spatial resolution (250 m–1 km) limits the applicability in monitoring fragmented small cropland patches [32]. The Landsat imagery is a suitable data source with 30 m spatial resolution [33]. However, due to its low temporal resolution (16 days), it is difficult to obtain enough cloudless images for time series analysis [2]. The wide-field view (WFV) images of GF-6 have a temporal resolution of 2 days, a spatial resolution of 16 m, a width of 800 km, and two red-edge bands, which can gather the advantages of most satellites. Therefore, it has great potential to effectively map rice in a large area of fragmented landscapes.

Chlorophyll content, leaf area index (LAI), and above-ground dry biomass (AGB) are important parameters for rice growth evaluation [34]. Traditional crop growth monitoring mostly relies on ground investigation, which is time-consuming, laborious, and costly. It is destructive and difficult to monitor the growth in a large area [35,36]. Remote sensing technology makes it possible to monitor crop growth and physiological parameters rapidly and nondestructively in large areas, providing important technical support for non-destructive monitoring of crop growth [34,36–38]. Many achievements have been made in crop growth monitoring by remote sensing, including empirical statistical methods and radiative transfer models [39]. The empirical statistical methods include VI models [40–42], principal component regression [43,44], backpropagation (BP) neural network [36], partial least squares regression (PLSR) [45,46], which are widely used for their simplicity and flexibility.

Parametric statistical approaches based on vegetation indices (VIs) are the simplest and most extensive estimation approaches [47]. At the pre-heading stage, VIs usually shows a strong relationship with crop biomass, so most scholars built the relationships between VIs and AGB to monitor rice growth [34]. For example, Tao H et al. used stepwise regression (SWR) and PLSR methods based on VIs, red-edge parameters, and their combinations to accurately estimate AGB and LAI [39]. Devia C A used seven VIs and their combinations to estimate rice biomass in the large area based on Unmanned Aerial Vehicle (UAV) with an average correlation of 0.76 [48]. Bai Y Y et al. used time-series NDVI and EVI to monitor crop growth [49]. However, due to the spectral saturation under high biomass conditions, this relationship is worse at the post heading stage [34]. For example, Zhou et al. found that the emergence of rice panicles increased the difficulty of LAI estimation at the late growth stage of rice [50]. Moreover, Sakamoto et al. found the same result in rice, that is, the color index performed well before the heading stage but was poor at the late heading stage [51]. In recent years, UAV with its high spatial and temporal resolutions has become a platform for remote sensing applications in precision agriculture [34,35]. Data collected by UAV play an important role in crop growth monitoring. The red-edge band can reflect the growth of crops [42]. It is rare to use a red-edge band to monitor rice growth. Therefore, based on UAV data, red-edge bands of GF-6 images were used to monitor the growth of rice in a large area.

Given the difficulty of mapping large area rice and monitoring rice growth in the fragmented and complex landscape in southern China, in this study, GF-6 time-series images and red-edge bands were employed to map rice and monitor rice growth. We proposed a red-edge decision tree (REDT) algorithm and phenological parameters. Meanwhile, the traditional NNE methods based on GF-1 and GF-6 were compared. Furthermore, rice growth monitoring was conducted using UAV and GF-6 images. Finally, the rice map and the rice growth map were generated.

## 2. Materials and Data Processing

### 2.1. Study Area

The study site was located at Jingzhou City, Hubei Province, China (29°26′–31°37′N, 111°15′–114°05′E). It is one of the most important agricultural production areas in Hubei Province. It belongs to the subtropical monsoon climate zone. The annual precipitation is about 1100–1300 mm, mainly from May to October. Jingzhou City takes two-season rotation as the main farming method, in which rape and wheat are mainly planted in winter. In summer, the paddy field is mainly planted with rice, while dry land is mainly planted with soybean and corn. The planting area of rice is the largest. Rice seeds are usually sown at the end of April, and rice seedling plants are transplanted to paddy rice fields between late May and early June. Flooding is usually carried out about two weeks before rice transplanting, and it is a key practice in rice agriculture. Rice plants are mature in mid-September and harvested at the end of October. Corn and soybean are planted from mid-to-late May and are mature in late August to early September. The growth period for corn and soybean is similar to that of rice. Other vegetation such as woodland usually has longer growing seasons than rice, corn, and soybean. Jingzhou City is an important rice planting area in the Yangtze River Basin, with a large planting area and fragmented block. There are large areas of croplands, water bodies, buildings, woodland, and other land cover types. The paddy rice fields in the study are distributed in a scattered manner (Figure 1). Therefore, the study site is of great significance for the study of mapping paddy rice in the fragmented block.

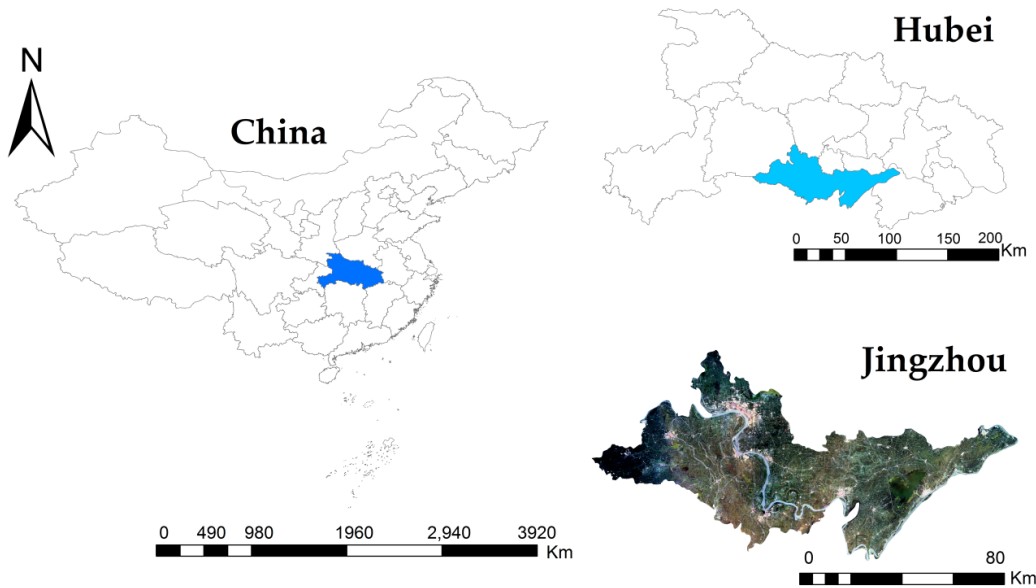

**Figure 1.** Location of the study site (29°26′–31°37′N, 111°15′–114°05′E).

*2.2. Data and Preprocessing*

2.2.1. GF-6/GF-1 Data and Preprocessing

Chinese GF-1 satellite is equipped with four WFV multispectral sensors (blue: 0.45–0.52 μm, green: 0.52–0.59 μm, red: 0.63–0.69 μm, and near-infrared: 0.77–0.89 μm), which can provide images with 800 km width, 4-day in temporal scale, and spatial resolution of 16 m. Compared with GF-1, GF-6 has added purple (0.40–0.45 μm), yellow (0.59–0.63 μm), red-edge band1 (0.69–0.73 μm), and red-edge band2 (0.73–0.77 μm). Moreover, the temporal resolution is shortened from 4 days to 2 days. A total of 24 GF-1 WFV images and nine GF-6 WFV images covering the study site were used in this study. All GF-1 and GF-6 data were obtained from the China center for resources satellite data and application (CCRSDA; http://www.cresda.com/CN/ (accessed on 1 February 2021)). The summary of the information on the application, sensors, acquisition date, number, temporal resolution, and central wavelength of these images is shown in Table 1.

**Table 1.** The information of GF-6 wide-field view (WFV) and GF-1 WFV.

| | Acquisition Date | Image Number | Sensor | Temporal Resolution | Central Wavelength/nm |
|---|---|---|---|---|---|
| GF-6 | 4 June 2019 | 2 | WFV | 2 days | B1 (blue) : 485 B2 (green): 555 B3 (red): 660 B4 (near-infrared): 830 B5 (red-edge1): 704 B6 (red-edge2): 752 B7 (coast blue): 425 B8 (yellow): 610 |
| | 27 July 2019 | 3 | | | |
| | 12 August 2019 | 2 | | | |
| | 20 August 2019 | 1 | | | |
| | 30 September 2019 | 2 | | | |
| GF-1 | 2 June 2019 | 4 | WFV1 WFV2 WFV3 WFV4 | 4 days | B1 (blue): 485 B2 (green): 555 B3 (red): 660 B4 (near-infrared): 830 |
| | 26 June 2019 | 4 | | | |
| | 2 August 2019 | 4 | | | |
| | 15 August 2019 | 4 | | | |
| | 12 September 2019 | 4 | | | |
| | 28 September 2019 | 4 | | | |

The preprocessing of GF-1 and GF-6 WFV images includes three steps: geometric correction, radiometric calibration, and atmospheric correction. The geometric correction was

conducted with the assistance of ASTER GDEM V2 data. For each image, radiometric calibration was processed using the Environment for Visualizing Images (ENVI) 5.3 software (Harris Geospatial Solutions, Inc.: Broomfield, CO, USA). The updating calibration parameters [52] were published in CCRSDA, obtained by a large number of calibration experiments in Chinese calibration fields. Atmospheric correction converts the radiance to reflectance and was performed using the Fast Line-of-Sight Atmospheric Analysis of Spectral Hypercubes (FLAASH) model in ENVI and executed using the Interactive Data Language (IDL) (Harris Geospatial Solutions, Inc.: Redlands, CA, USA). The related FLAASH parameters were obtained according to the acquisition time and imaging conditions. For all selected images, the cloud cover was less than 10%, almost cloudless. According to the acquisition time and imaging conditions, the relevant FLAASH parameters were obtained.

### 2.2.2. UAV Image Data and Preprocessing

In this study, an UAV (eight rotors, North Tiantu aviation technology development Co., Ltd, Beijing, China) (Figure 2c) equipped with a Mini-MCA system was used to obtain the multispectral images of the rice field (Ezhou City, Hubei Province, Figure 2a). The Mini-MCA system consisted of an array of twelve individual miniature digital cameras (Mini-MCA 12, Tetracam, Inc., Chatsworth, CA, USA) (Figure 2d). The Mini-MCA 12 was mounted on a DJIS1000 octocopter (SZ DJI Technology Co., Ltd., Shenzhen, China). Each camera was equipped with a customer-specified bandpass filter centered at the wavelength of 490 nm, 520 nm, 550 nm, 570 nm, 670 nm, 680 nm, 700 nm, 720 nm, 800 nm, 850 nm, 900 nm, and 950 nm, respectively. These spectral bands range from visible to near-infrared and can be used to analyze vegetation growth parameters [53]. The Mini-MCA system was fixed in the UAV by a gimbal which can help to compensate for the UAV movement (pitch and roll) during the flight and guarantee close to nadir image collection [34]. The UAV flight was conducted in clear skies between 10 am and 2 pm local time, when the change of solar zenith angle was minimal and the illumination was stable. The ground sample distance (GSD) of the UAV image is 5.5 cm. The altitude and speed of each flight were 100 m and 4 m/s, respectively, covering the whole rice field. Data were obtained every five days from 26 June 2019 to 3 September 2019.

Because the MCA system has a significant effect on camera registration error, band-to-band registration is carried out in the laboratory before the flight. So, the corresponding pixels of each lens can be overlapped in space on the same focal plane. The empirical linear calibration model was used for calibration, and the image digital number (DN) was converted into the surface reflectance (R) [54,55]. Eight calibration ground targets approximate Lambertian, which have been calibrated in the laboratory, were spread on the flat and unshaded area of the experimental field. The reflectance of the calibration targets is 0.03, 0.06, 0.12, 0.24, 0.36, 0.48, 0.56, and 0.80, respectively. Based on the linear relationship between DN and R, the reflectance value was obtained

$$R_\lambda = DN_\lambda \times G_\lambda + B_\lambda \tag{1}$$

where $\lambda$ was the band wavelength of MCA camera; $R_\lambda$ and $DN_\lambda$ were the surface reflectance and the digital number of a pixel at wavelength $\lambda$, respectively; $G_\lambda$ and $B_\lambda$ were gains and bias at wavelength $\lambda$, respectively. For each wavelength $\lambda$, $G_\lambda$ and $B_\lambda$ can be calculated using the least-square method by R and DN values (referring to $DN_{0.03}$, $DN_{0.06}$, $DN_{0.12}$, $DN_{0.24}$, $DN_{0.36}$, $DN_{0.48}$, $DN_{0.56}$, and $DN_{0.80}$) of eight calibration targets.

$$
\begin{pmatrix} 0.03 \\ 0.06 \\ 0.12 \\ 0.24 \\ 0.36 \\ 0.48 \\ 0.56 \\ 0.80 \end{pmatrix} = \begin{pmatrix} DN_{0.03} \\ DN_{0.06} \\ DN_{0.12} \\ DN_{0.24} \\ DN_{0.36} \\ DN_{0.48} \\ DN_{0.56} \\ DN_{0.80} \end{pmatrix} \times G_\lambda + B \tag{2}
$$

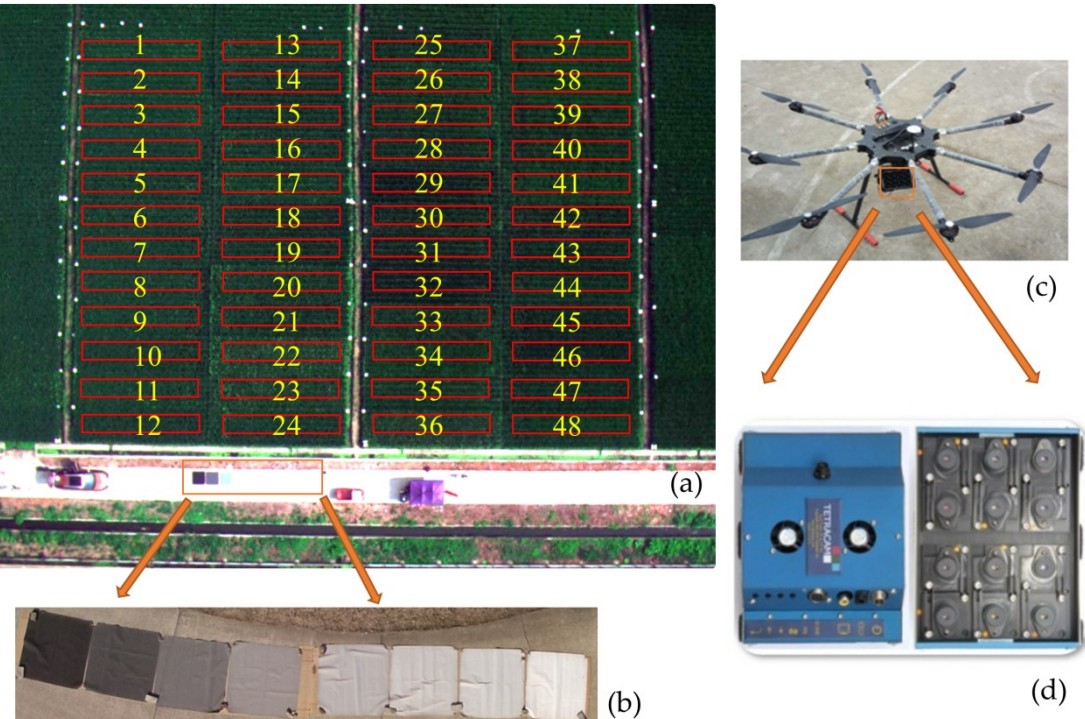

**Figure 2.** The illustration of (**a**) rice field photographed by UAV, 48 rice plots (**b**) eight calibration ground targets (**c**) UAV (**d**) Mini-MCA.

### 2.3. Ancillary Data

The ASTER GDEM V2 data, used for the geometric correction of satellite images, is a global digital elevation model (DEM) with 30 m resolution. The sample collection is divided into two ways: field measurement with GPS (Figure 3) and visual interpretation of Google Earth. 1300 pixels of rice samples and 800 pixels of non-rice samples were selected in the field from June to October 2019. The acquisition time of all samples is consistent with the image acquisition time. Google Earth (GE) provides free high-resolution images, which can help us with the accurate visual interpretation of features. These sample data points were geo-referenced to remote sensing images by latitude and longitude coordinates. A total of 2197 samples (137,841pixels) were obtained, which were distributed in Jianli County, Gongan County, Jiangling County, Jingzhou City District, and Honghu county. These samples were labeled into six categories, rice, corn, soybean, water, woodland, and building.

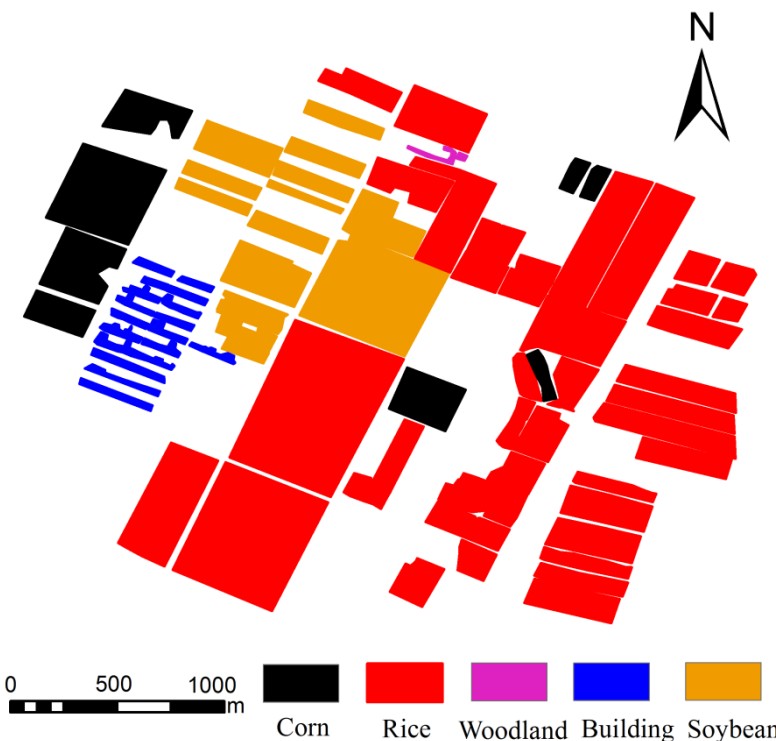

**Figure 3.** Field measured data labeled plot (30°15′56″–30°16′57″N, 112°13′33″–112°14′54″E).

There were 48 representative rice plots (Figure 2a), and the plots were of the same size about 60 m². There are 48 rice hybrid species recommended by breeding experts. All rice cultivars were sown on 11 May 2019 and transplanted on 8 June 2019 with the transplanting density of 15,000 plants/ha. The planting density, nitrogen rate, and water rate of these plots were the same. A protected planting area with a width of 2 m was provided around the rice plot. From 26 June 2019 to 3 September 2019, destructive samplings were conducted every five days, and a total of 13 times were carried out for each plot during the whole growing period. The sampling time is consistent with the flight time of UAV. For each plot, three bundles were randomly dug out from soil with root, placed in a bucket full of water, and taken to the laboratory. The root of the plant was cut, the remaining part of the plant was cleaned and divided into stems, leaves, and ears. All the samples were dried for half an hour at 105 °C and later dried at 80 °C until their weights remained unchanged. All samples were weighed and recorded, and the AGB (g·m⁻²) per square meter was calculated.

### 3. Methodology

The time series- and phenology-based paddy rice mapping methodology (Figure 4) mainly involved the following steps: (1) Non-cropland masking by using images after pre-processing; (2) Calculate the time series VIs (NDWI, NDVI, NDRE, Normalized Red edge Difference Index (NREDI)) and the red-edge integral value; (3) Mapping paddy rice fields by using the REDT method and compared with NNE method; (4) Accuracy Assessment. (5) Rice growth monitoring by using time-series images.

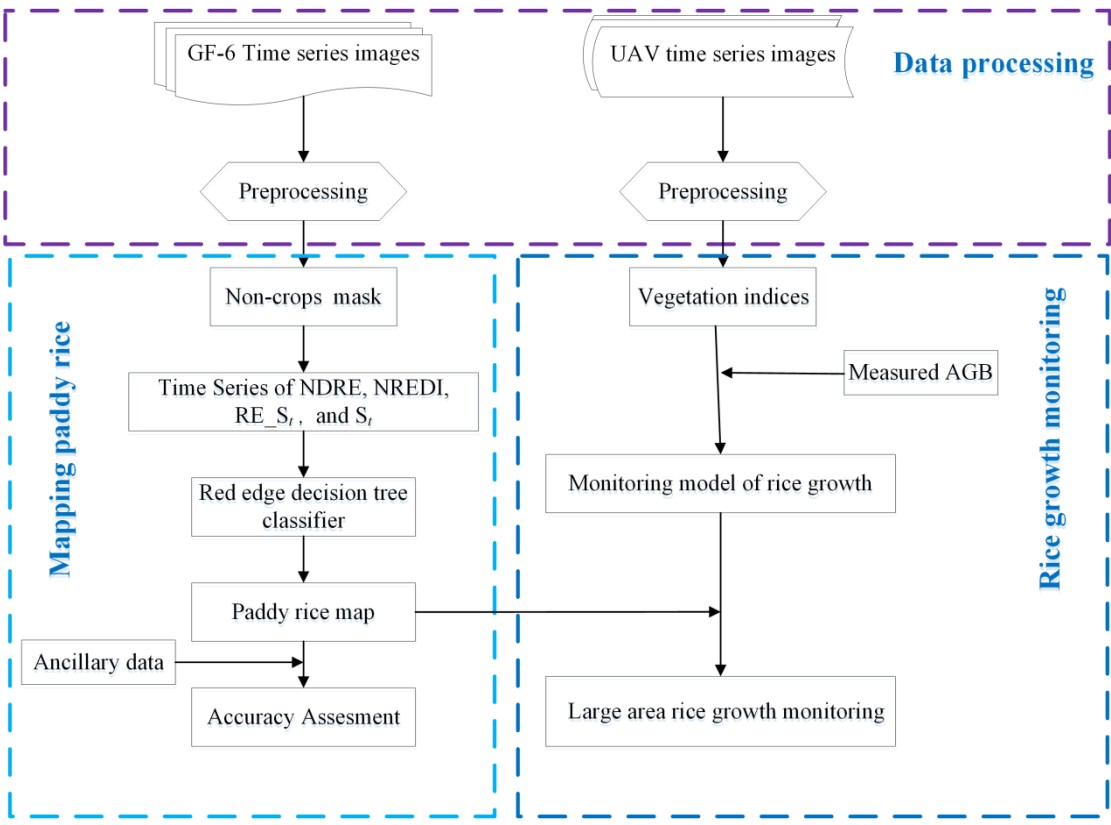

**Figure 4.** The flowchart of the methodology.

### 3.1. Non-Cropland Masks

The paddy fields are a mixture of water, soil, and rice seedlings in the paddy rice transplanting period. It is a unique feature of rice. Moreover, some noises could be caused by non-cropland, such as bare land and buildings due to some unexpected weather effects. Based on the previous research results [9] and combined with GF-6 data, the non-cropland masking steps are as follows: The water is masked by setting NDVI < 0, NDWI > 0 (the image of 27 July), and the building is masked by setting NDVI < 0.45 (the image of 27 July), NDVI < 0.4 (the image of 12 August). The woodland is masked by setting NDVI > 0.6 (the image of 4 June), NDVI > 0.4 (the image of 30 September). The time series VIs were calculated by the average value of several pixels obtained from field measurement and visual interpretation based on GF-6 images and GF-1 images [21]. The formula is as follows:

$$\text{NDWI} = \frac{R_{555nm} - R_{830nm}}{R_{555nm} + R_{830nm}} \tag{3}$$

$$\text{NDVI} = \frac{R_{830nm} - R_{660nm}}{R_{830nm} + R_{660nm}} \tag{4}$$

$$\text{EVI2} = 2.5 * \frac{R_{830nm} - R_{660nm}}{R_{830nm} + 2.4 * R_{660nm} + 1} \tag{5}$$

In which, $R_{555nm}$, $R_{660nm}$, and $R_{830nm}$ are the reflectance corresponding to the central wavelength of 550 nm, 660 nm, and 830 nm, respectively.

### 3.2. Red-Edge Decision Tree Method Based on Time Series

#### 3.2.1. Phenological Analysis of Rice, Corn, and Soybean

The main crop types in the study area from May to October are rice, corn, and soybean. Their specific phenological calendars are shown in Figure 5. The red vertical lines in Figure 5 represents the date of the image (GF-6) used in our study. The growth period of corn and soybean is slightly shorter, from May to the end of August. They have many phenological stages during the same period, but their phenological differences are concentrated in July and August. The growth period of rice is the longest and has unique characteristics in the transplanting period.

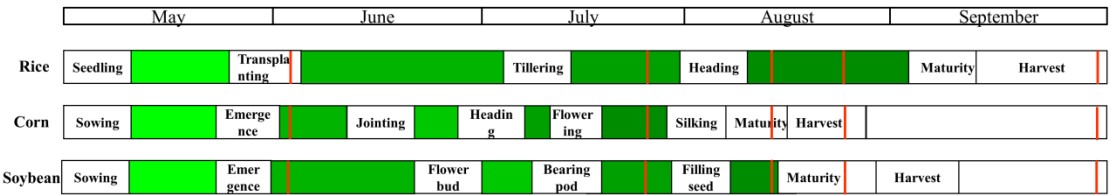

**Figure 5.** Phenological calendars for the main crop types, and the dates of the selected GF-6 images.

#### 3.2.2. Red-Edge Decision Tree Classification

In this paper, a method of spectral integration from the red-edge band1 to the near-infrared band (the area enclosed from the red-edge band1 to the near-infrared band) using time series images is proposed. At the same time, the time series phenological parameters based on the red-edge band (NDRE, NREDI) are calculated. A red-edge vegetation index integration method based on time series is proposed. These can be used to construct the red-edge decision tree (REDT) for rice mapping. The functions are shown in Formulas (6)–(10).

$$RE\_S_t = \frac{(R_{752nm} + R_{704nm})(752 - 704) + (R_{830nm} + R_{752nm})(830 - 752)}{2} \tag{6}$$

$$S_t = \frac{(R_{660nm} + R_{830nm})(830 - 660)}{2} \tag{7}$$

$$VI\_S_{t_i \sim t_j} = \frac{(VI_j + VI_i)(t_j - t_i)}{2} \tag{8}$$

$$NDRE = \frac{R_{830nm} - R_{704nm}}{R_{830nm} + R_{704nm}} \tag{9}$$

$$NREDI = \frac{R_{752nm} - R_{704nm}}{R_{752nm} + R_{704nm}} \tag{10}$$

where $RE\_S_t$ represents the spectral integral value of the image at time t from the central wavelength of 704 nm to the central wavelength of 830 nm. $S_t$ represents the spectral integral value of the image at time t from the central wavelength of 660 nm to the central wavelength of 830 nm. $VI\_S_{ti \sim tj}$ represents the integral value of vegetation index (VI) from time $t_i$ to $t_j$, $VI_i$, $VI_j$ represents the vegetation index values corresponding to $t_i$ and $t_j$, respectively. $R_{704nm}$ and $R_{752nm}$ are the reflectance corresponding to the central wavelength of 704 nm and 752 nm, respectively.

### 3.3. Methods of the NDVI, EVI, and NDWI Time Series

For comparison purposes, we also used the NNE method to highlight the advantages of the red-edge band in rice mapping. Since rice fields are a mixture of water, soil, rice seedlings in the transplanting stage, and covered by rice at the heading stage, the NDWI value of rice is quite different from that of other crops. Combined with previous research [21] and GF-1/GF-6 images, the condition of the NNE method based on GF-6 by using histogram threshold method is set as $NDVI\_S_{12\ August \sim 20\ August} < 4.3$, $EVI\_S_{12\ August \sim 20\ August} < 3$ (corn is masked), $NDWI_{04\ June} - NDWI_{20\ August} < 0.4$, $S_{27\ July} < 32$ (soybean is masked)

(Figure 6a). The condition of the NNE method based on GF-1 by using histogram threshold is set as $S_{15\ August} < 32$, $EVI\_S_{02\ August\sim12\ September} < 15$ (corn is masked), $NDWI_{02\ June} - NDWI_{15\ August} < 0.35$, $NDVI\_S_{02\ August\sim12\ September} < 23$, $EVI\_S_{02\ August\sim12\ September} < 20$ soybean is masked)(Figure 6b).

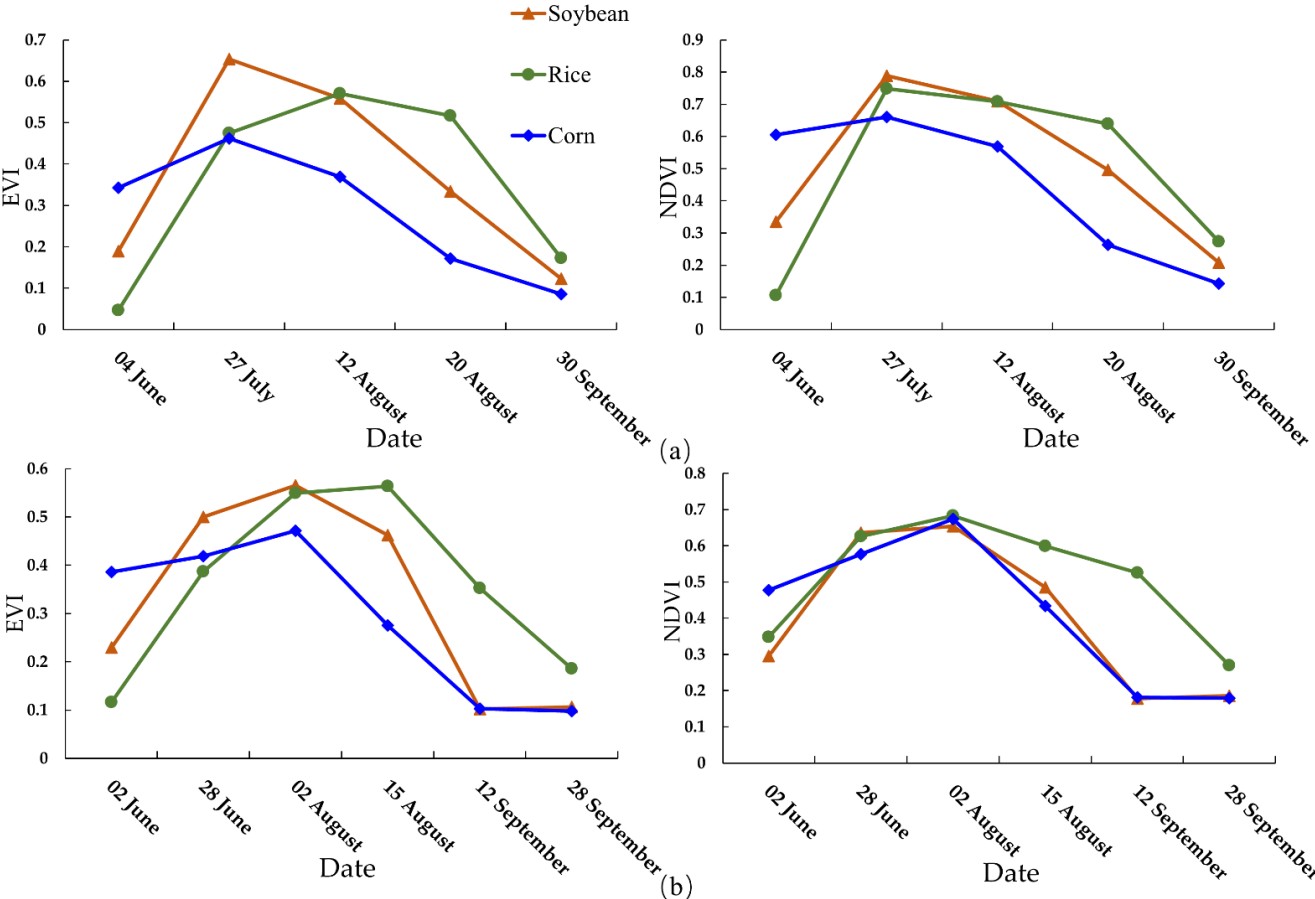

**Figure 6.** Normalized difference vegetation index (NDVI), enhanced vegetation index (EVI) time series of rice, corn, and soybean-based on GF-6 (**a**) and GF-1 (**b**).

### 3.4. Results Validation

The results of paddy rice mapping were validated using the confusion matrix. Overall accuracy, kappa coefficient, user accuracy, and producer accuracy were used to assess the accuracy of the REDT method. The validating samples of results are introduced in Section 2.3.

### 3.5. Rice Growth Monitoring Based on Red-Edge Band

Based on the change of red-edge band in different growth stages of rice in Section 3.2, two red-edge bands of GF-6 were applied to monitor the growth of rice. The VIs used are shown in Table 2.

**Table 2.** The vegetation index for rice growth monitoring based on UAV and GF-6 images.

| | VIs | Formula | Reference |
|---|---|---|---|
| UAV | Normalized Difference Vegetation Index (NDVI) | $\frac{R_{830nm}-R_{704nm}}{R_{704nm}-R_{660nm}}$ | Rouse et al., 1974 [56] |
| | Normalized Difference Red edge (NDRE) | $\frac{R_{800nm}-R_{720nm}}{R_{800nm}+R_{720nm}}$ | Glenn et al., 2010 [57] |
| | MERIS Terrestrial Chlorophyll Index (MTCI) | $\frac{R_{800nm}-R_{720nm}}{R_{720nm}-R_{670nm}}$ | Dashand Curran, 2004 [58] |
| | Red-edge Chlorophyll Index (CI$_{red\text{-}edge}$) | $\frac{R_{800nm}}{R_{720nm}}-1$ | Gitelson et al., 2005 [59] |
| | Two-band Enhanced Vegetation Index (EVI2) | $\frac{2.5(R_{800nm}-R_{670nm})}{R_{800nm}+2.4R_{670nm}+1}$ | Jiang et al., 2008 [60] |
| | Leaf Chlorophyll Index (LCI) | $\frac{R_{850nm}-R_{720nm}}{R_{850nm}-R_{680nm}}$ | Datt, B. et al, 1999 [61] |
| | Normalized Red-edge Difference Index (NREDI) | $\frac{R_{720nm}-R_{700nm}}{R_{720nm}+R_{700nm}}$ | Feng G U, 2019 [62] |
| GF-6 | Normalized Difference Vegetation Index (NDVI) | $\frac{R_{830nm}-R_{660nm}}{R_{830nm}+R_{660nm}}$ | Rouse et al., 1974 [56] |
| | Normalized Difference Red edge (NDRE$_{red\text{-}edge5}$) | $\frac{R_{830nm}-R_{704nm}}{R_{830nm}+R_{704nm}}$ | Glenn et al., 2010 [57] |
| | Normalized Difference Red edge (NDRE$_{red\text{-}edge6}$) | $\frac{R_{830nm}-R_{752nm}}{R_{830nm}+R_{752nm}}$ | Dashand Curran, 2004 [58] |
| | MERIS Terrestrial Chlorophyll Index (MTCI$_{red\text{-}edge5}$) | $\frac{R_{830nm}-R_{704nm}}{R_{704nm}-R_{660nm}}$ | Dashand Curran, 2004 [58] |
| | MERIS Terrestrial Chlorophyll Index (MTCI$_{red\text{-}edge6}$) | $\frac{R_{830nm}-R_{752nm}}{R_{752nm}-R_{660nm}}$ | Gitelson et al., 2005 [59] |
| | Red-edge Chlorophyll Index (CI$_{red\text{-}edge5}$) | $\frac{R_{830nm}}{R_{704nm}}-1$ | Gitelson et al., 2005 [59] |
| | Red-edge Chlorophyll Index (CI$_{red\text{-}edge6}$) | $\frac{R_{830nm}}{R_{752nm}}-1$ | Jiang et al., 2008 [60] |
| | Two-band Enhanced Vegetation Index (EVI2$_{red\text{-}edge5}$) | $\frac{2.5(R_{830nm}-R_{704nm})}{R_{830nm}+2.4R_{704nm}+1}$ | Jiang et al., 2008 [60] |
| | Two-band Enhanced Vegetation Index (EVI2$_{red\text{-}edge6}$) | $\frac{2.5(R_{830nm}-R_{752nm})}{R_{830nm}+2.4R_{752nm}+1}$ | Feng G U, 2019 [62] |
| | Red-edge Triangle Chlorophyll Index (TCI$_{red\text{-}edge5}$) | $1.2(R_{704nm}-R_{555nm})-1.5(R_{660nm}-R_{555nm})\sqrt{\frac{R_{704nm}}{R_{660nm}}}$ | Feng G U, 2019 [62] |
| | Red-edge Triangle Chlorophyll Index (TCI$_{red\text{-}edge6}$) | $1.2(R_{704nm}-R_{555nm})-1.5(R_{660nm}-R_{555nm})\sqrt{\frac{R_{752nm}}{R_{660nm}}}$ | Feng G U, 2019 [62] |
| | Red-edge Transformation Chlorophyll Absorption Reflectance Index (TCARI$_{red\text{-}edge5}$) | $3(R_{704nm}-R_{660nm})-0.2(R_{704nm}-R_{555nm})\frac{R_{704nm}}{R_{555nm}}$ | Feng G U, 2019 [62] |
| | Red-edge Transformation Chlorophyll Absorption Reflectance Index (TCARI$_{red\text{-}edge6}$) | $3(R_{752nm}-R_{660nm})-0.2(R_{752nm}-R_{555nm})\frac{R_{752nm}}{R_{555nm}}$ | Feng G U, 2019 [62] |

## 4. Results

### 4.1. Analysis of Red-Edge Characteristics of Different Crops

By calculating the average values of abundant pixels obtained from field measurements, we have drawn the spectral curves of soybean, corn, and rice in different periods (24 June, 27 July, 12 August, 20 August, 30 September), as shown in Figure 7a. The reflectance of soybean from the red-edge1 band to the red-edge2 band changed greatly on 27 July, which was significantly larger than that of rice and corn. On 20 August, soybean was mature and stopped growing, and the spectral reflectance of the red-edge band and near-infrared band decreased. However, rice is at the heading stage, and it grows vigorously, and the reflectance of the red-edge band and the near-infrared band is increasing. So, they show the opposite changing trend in the red edge band and near-infrared band. Moreover, the change rate of rice from the red-edge1 to red-edge2 was larger than that of soybean and corn. During this period, the corn is in the harvest period, and the spectral curve is close to the soil spectral curve. On 30 September, corn, soybean, and rice were all harvested, so the spectral curves were close to the soil spectral curves. On 27 July, 12 August and 20 August, the growth of rice, soybean, and corn changed greatly, especially in the red-edge band. The images of these three periods are very important for identifying rice, corn, and soybean.

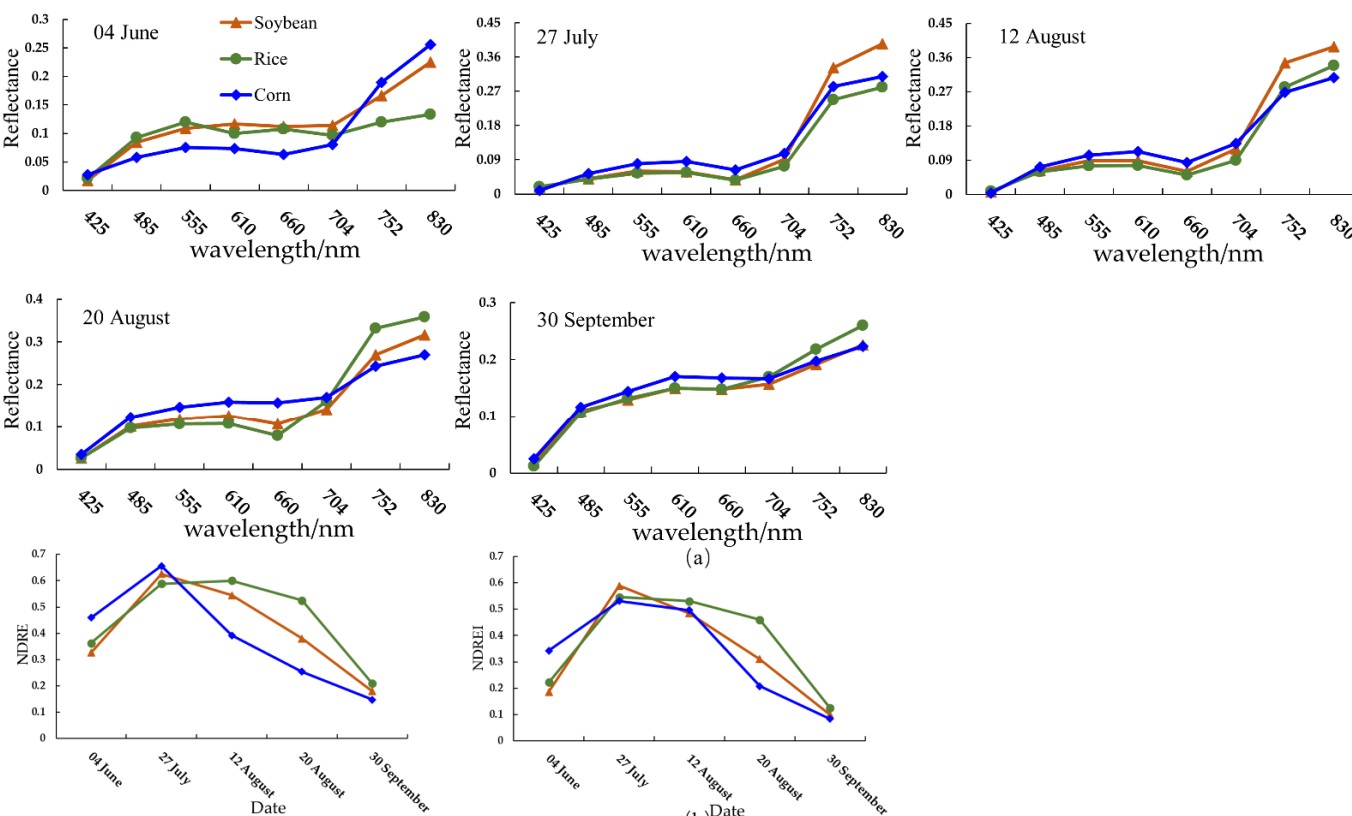

**Figure 7.** The spectral curves of soybean, corn, and rice in different periods (**a**) and Normalized Red-edge Difference Index (NREDI) time series and NDRE time series (**b**), the dates are 4 June 2019, 27 July 2019, 12 August 2019, 30 September 2019 respectively.

As shown in Figure 7, 27 July, 12 August, 20 August, the red-edge changes of rice, soybean, and corn in these three periods are quite different. We randomly selected 270 pixels of rice samples, 220 pixels of soybean samples, 120 pixels of corn samples, and used the histogram threshold method to distinguish them (Figure 8). The soybean was masked by setting conditions $RE\_S_{27\ July} > 35$, $RE\_S_{12\ August} > 38$ and the corn was masked by setting conditions $RE\_S_{20\ August} < 34$, $NDRE\_S_{12\ August\sim20\ August} < 3.6$, $NDREI\_S_{12\ August\sim20\ August} < 3.1$. The remaining crop is rice through the implementation of the above process.

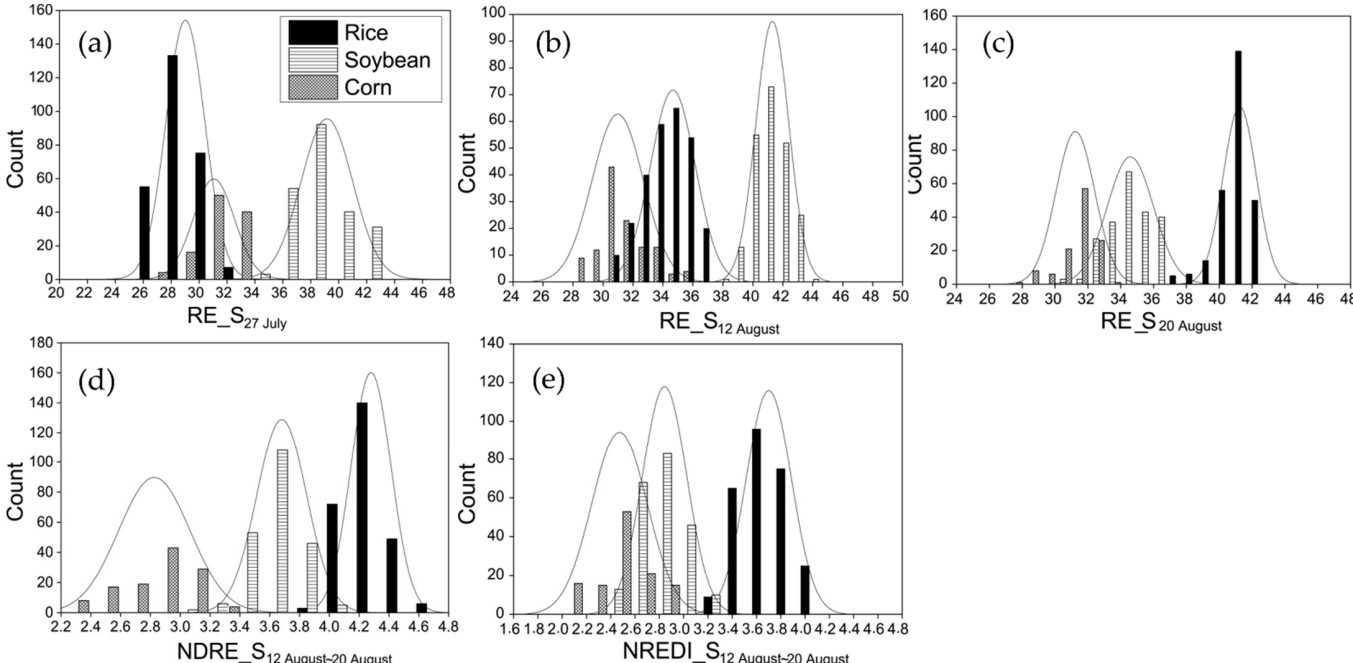

**Figure 8.** Threshold histogram of integral value from red-edge band to near-infrared band on (**a**) 27 July 2019; (**b**) 12 August 2019; (**c**) 20 August 2019; and the integral value of red-edge vegetation index based on time series (**d**) $NDRE\_S_{12\ August\sim20\ August}$ (**e**) $NDREI\_S_{12\ August\sim20\ August}$ for three croplands.

### 4.2. Mapping Paddy Rice Using REDT Method and Accuracy Assessment

Combined with the time series images and the change characteristics of red-edge bands in different periods of rice, the red-edge spectral integral and red-edge vegetation index integral were constructed to distinguish rice from other crops. The rice mapping results of the local area A (30°15′56″–30°16′57″N, 112°13′33″–112°14′54″E) and B (29°59′8″–29°59′33″N, 112°12′51″–112°13′33″E) are shown in Figure 9, and the accuracy of three maps are presented in Table 3. The confusion matrix and kappa coefficient were used to evaluate the accuracy. The overall accuracy of the REDT method is 93.85% (A), 94.06% (B) and the kappa coefficient is 0.87 and 0.89, respectively. Compared with the NNE method using GF-6 and GF-1 images, the overall accuracy was improved by about 7% and the kappa coefficient was improved by about 0.15, which showed that the red-edge band had a great contribution to distinguish rice from other crops. The REDT method was effective in distinguishing rice, corn, and soybean.

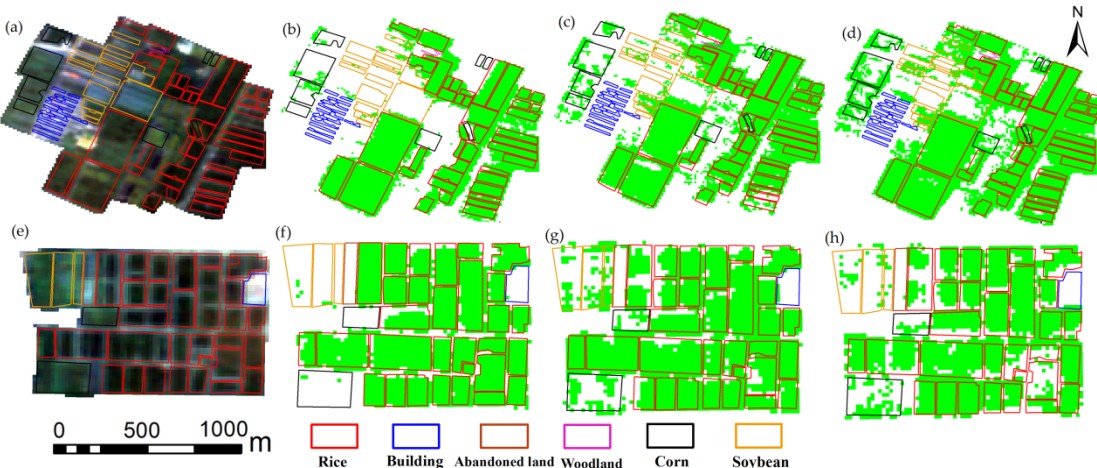

**Figure 9.** The comparison of three local paddy rice maps in A and B region ((**a**) A-GF-6 image; (**b**) A-red-edge decision tree (REDT) method based on GF-6 rice map; (**c**) A-normalized difference vegetation index (NDVI), normalized difference water index (NDWI), enhanced vegetation index (NNE) method based on GF-6 rice map; (**d**) A-NNE method based on GF-1 rice map; (**e**) B-GF-6 image; (**f**) B-REDT method based on GF-6 rice map; (**g**) B-NNE method based on GF-6 rice map; (**h**) B-NNE method based on GF-1 rice map).

**Table 3.** Confusion matrix of local (A and B) and Jingzhou city accuracy assessment using field measured data and Google Earth images (A: 800 pixels rice samples, 1300 pixels non-rice samples; B: 900 pixels rice samples, 700 pixels non-rice samples; Jingzhou city: 4000 pixels rice samples, 6000 pixels non-rice samples).

| Study Area | Paddy Rice Map | Class | PA% | UA% | OA% | Kappa Coefficient |
|---|---|---|---|---|---|---|
| A | GF-6-REDT | Rice | 91.85 | 93.04 | 93.85 | 0.87 |
| | | Non-rice | 95.24 | 94.40 | | |
| | GF-6-NNE | Rice | 85.82 | 81.48 | 86.42 | 0.72 |
| | | Non-rice | 86.84 | 90.07 | | |
| | GF-1-NNE | Rice | 86.37 | 77.55 | 85.29 | 0.69 |
| | | Non-rice | 84.61 | 90.98 | | |
| B | GF-6-REDT | Rice | 94.80 | 94.48 | 94.06 | 0.89 |
| | | Non-rice | 93.14 | 93.54 | | |
| | GF-6-NNE | Rice | 89.09 | 87.78 | 87.31 | 0.74 |
| | | Non-rice | 85.19 | 86.73 | | |
| | GF-1-NNE | Rice | 90.73 | 84.76 | 86.37 | 0.72 |
| | | Non-rice | 81.42 | 88.52 | | |
| Jingzhou City | GF-6-REDT | Rice | 90.01 | 89.60 | 91.10 | 0.82 |
| | | Non-rice | 91.93 | 92.26 | | |
| | GF-6-NNE | Rice | 81.76 | 78.11 | 83.39 | 0.66 |
| | | Non-rice | 84.49 | 87.26 | | |
| | GF-1-NNE | Rice | 86.34 | 75.43 | 82.17 | 0.64 |
| | | Non-rice | 79.06 | 88.60 | | |

At the same time, the REDT method is applied to Jingzhou City, and the result is shown in Figure 10a. The results of the NNE method based on GF-6 and GF-1 are shown in Figure 10b,c. The accuracy of the REDT method is still higher than that of the NNE method, and the overall accuracy is 91.10%, kappa coefficient is 0.82. Compared with the local area A and B, the accuracy is slightly reduced. However, the accuracy of the traditional NNE method in Jingzhou City is significantly lower than that in local areas A and B (Table 3). The



overall accuracy and kappa coefficient are 83.39% (GF-6), 82.17% (GF-1) and 0.66 (GF-6), 0.64 (GF-1), respectively, and the effect is poor. Based on GF-6 images, the accuracy of the REDT method is improved by about 8%, and the kappa coefficient is improved by about 0.16, which proves the importance of the red-edge band in rice mapping. The user accuracy of the NNE method is relatively worse in both local and large areas, because of the strong confusion of rice, corn, and soybean at these phenological stages. The proposed method demonstrated its effectiveness in rice mapping under fragmented landscape conditions, and the results proved the effectiveness and superiority of this method in rice mapping.

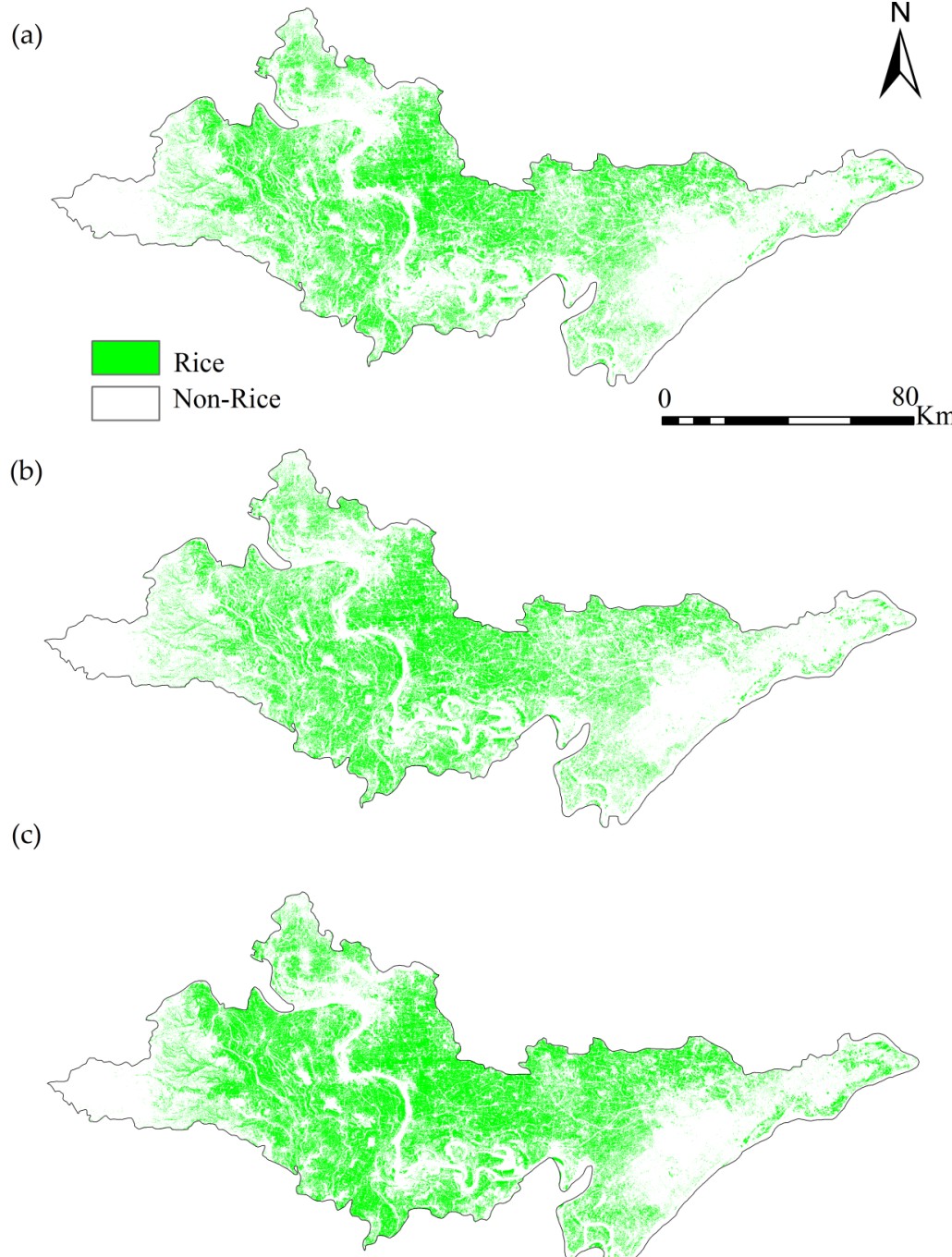

**Figure 10.** The rice mapping of Jingzhou City ((**a**) REDT method based on GF-6 rice map; (**b**) NNE method based on GF-6 rice map; (**c**)-NNE method based on GF-1 rice map).

### 4.3. Adaptability Verification of REDT Method

To verify the applicability of the REDT method, Tianmen City, Hubei Province was selected as the validation area. Figure 11 shows the mapping results of rice in the local area (30°26′4″–30°28′32″N, 113°21′59″–113°24′4″E) of Tianmen City. The Confusion matrices of the three maps are presented in Table 4. Figure 11b shows the REDT method can identify the broken rice field completely and is very effective for identifying rice in the fragmented landscape in southern China. The overall accuracy and kappa coefficient of REDT are 93.04% and 0.85, respectively (Table 4), which is visibly better than the traditional NNE method and is consistent with the conclusion obtained in Section 4.1. The REDT method is better than the NNE method in the identification of fragmented landscape, with less noise and more complete for the identification of the rice field.

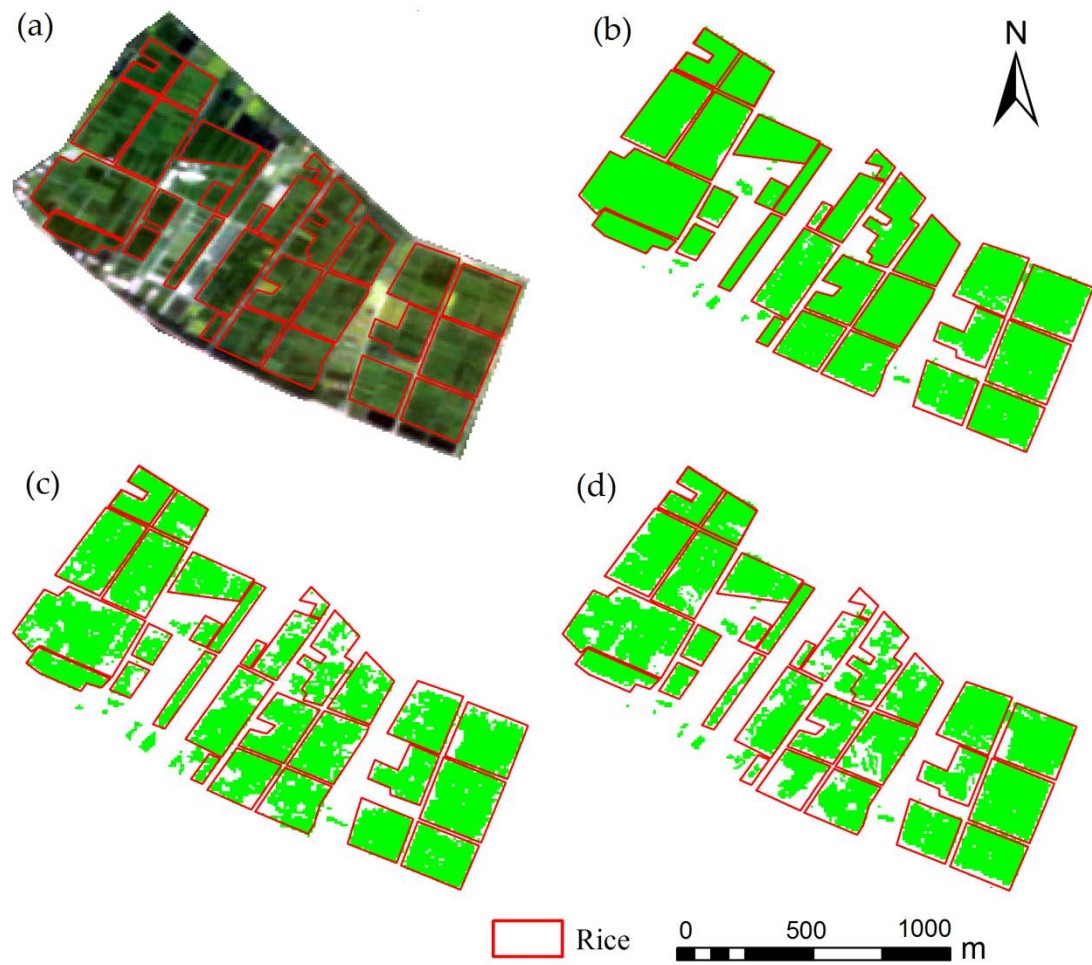

**Figure 11.** The comparison of three local paddy rice maps of Tianmen city ((**a**) GF-6 image; (**b**) REDT method based on GF-6 rice map; (**c**) NNE method based on GF-6 rice map; (**d**) NNE method based on GF-6 rice map).

**Table 4.** Confusion matrix of local accuracy assessment using Google Earth images.

| Paddy Rice Map | Class | PA (%) | UA (%) | OA (%) | Kappa Coefficient |
|---|---|---|---|---|---|
| GF-6-REDT | Rice | 94.55 | 93.86 | 93.04 | 0.85 |
| | Non-rice | 90.81 | 91.80 | | |
| GF-6-NNE | Rice | 87.68 | 90.64 | 87.17 | 0.73 |
| | Non-rice | 86.41 | 82.38 | | |
| GF-1-NNE | Rice | 85.26 | 91.04 | 86.08 | 0.71 |
| | Non-rice | 87.33 | 79.68 | | |

### 4.4. Rice Growth Monitoring

In the above mapping rice research, we conclude that the two red-edge bands played an important role in the rice growth stage. In this paper, the AGB is used to represent the growth of rice, and a suitable model is established through the measured AGB and UAV data. The UAV data contains two red-edge bands, the central wavelengths are 700 nm and 720 nm (with 10 nm bandwidth), and the GF-6 data has two red-edge bands with central wavelengths of 704 nm and 752 nm (with 40 nm bandwidth). The wavelength of 700 nm (UAV data) is similar to that of 704 nm (GF-6 data). The 704 nm band of the GF-6 image can replace the 700 nm band of UAV data. At the pre-heading stage, the fit plot of seven VIs and AGB is shown in Figure 12. MTCI performs best with a $R^2$ value of 0.90. Therefore, we can apply this vegetation index model to the GF-6 image (pre-heading stage), and use this model to get simulated AGB of rice. The red-edge vegetation index of rice calculated by the GF-6 image is fitted with the simulated AGB, and the results are shown in Figure 13.

In the post-heading stage, the fitting results of the vegetation index calculated by UAV data and AGB are worse (Figure 14), and the $R^2$ value ranged from 0.56 to 0.62. The single vegetation index model cannot estimate AGB well. Therefore, we use the PLSR model combined with multiple VIs, in which 280 sets of data were used for modeling and 152 sets of data were used for verification. The fitting results between the measured AGB and estimated AGB are shown in Figure 15, and the $R^2$ was 0.82, which is much better than the single vegetation index model. The model was applied to GF-6 images (post heading stage), and the simulated AGB was obtained. The red edge vegetation index of rice calculated by the GF-6 image is fitted with the simulated AGB, and the results are shown in Figure 16. NREDI performed best with a $R^2$ value of 0.84. The CI, NDRE, and EVI calculated by the red edge1 band are better than those calculated by the red edge2 band. The TCI and TCARI calculated by the red edge2 band are better than those calculated by the red edge1 band. This result is the same as that obtained at the pre-heading stage.

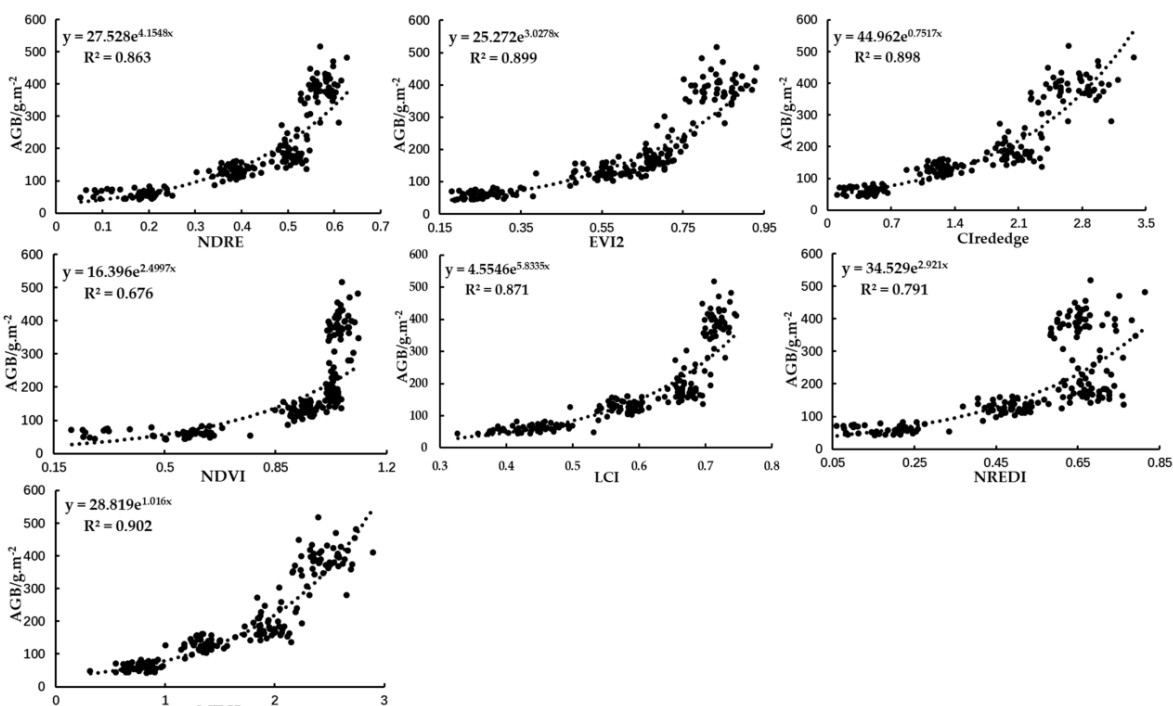

**Figure 12.** The fitting results of vegetation index and above-ground dry biomass (AGB) based on UAV at the pre-heading stage.

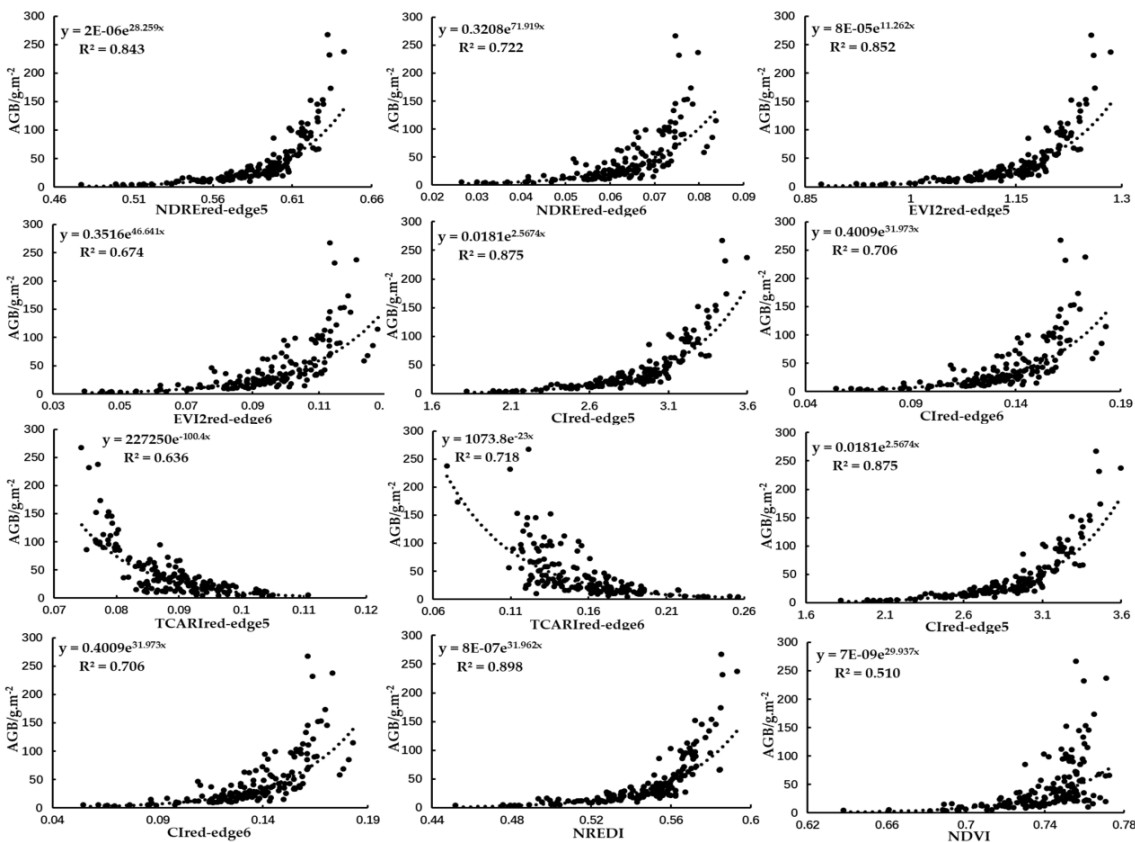

**Figure 13.** The fitting results of red-edge vegetation index and simulated AGB based on GF-6 images at the pre-heading stage.

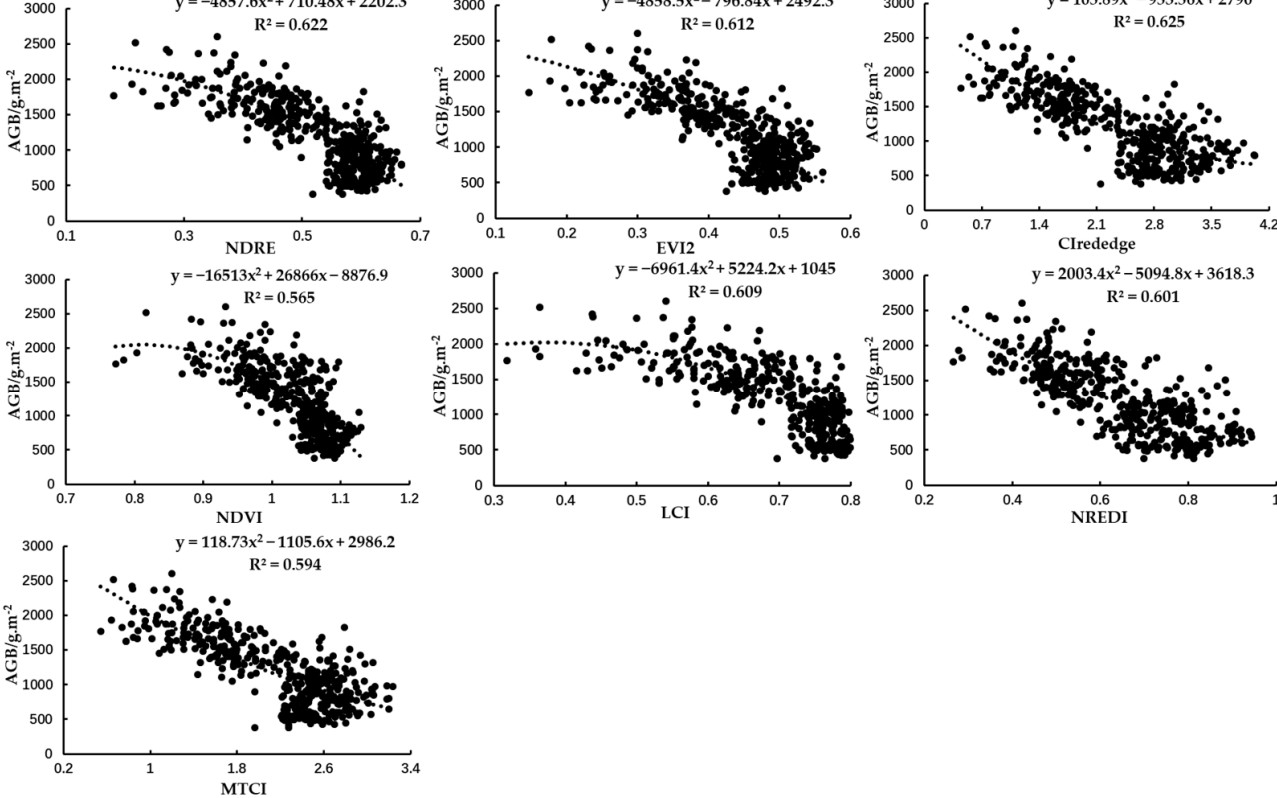

**Figure 14.** The fitting results of vegetation index and AGB based on UAV in the post-heading stage.

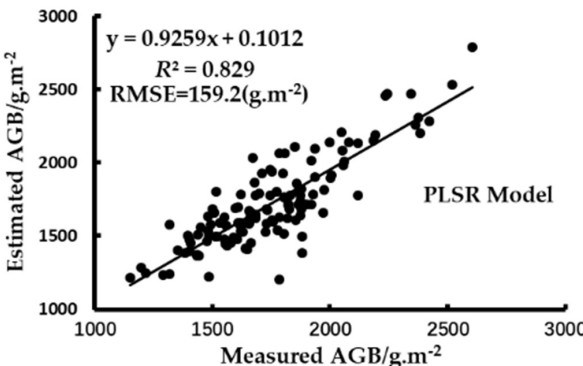

**Figure 15.** The fitting results of measured AGB and estimated AGB based on the partial least squares regression (PLSR) model.

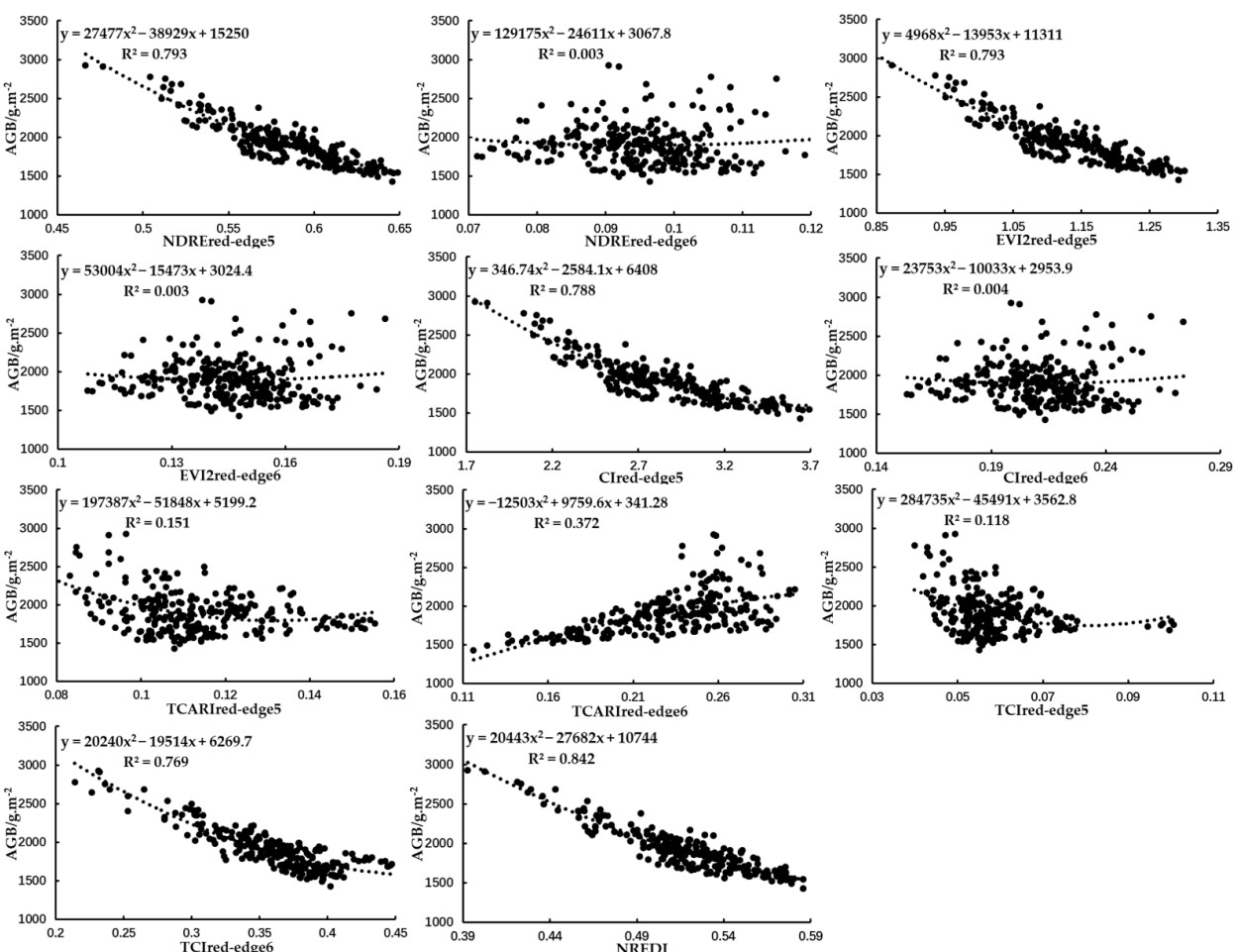

**Figure 16.** The fitting results of red-edge vegetation index and simulated AGB based on GF-6 images.

## 5. Discussion

### 5.1. Advantages of REDT Method in Rice Mapping Strategy

Most scholars have used time series vegetation index (NDVI, NDWI, EVI) and phenological characteristics for mapping rice [2,8,9,21,30]. However, in the vigorous growth stage, the supersaturation of vegetation index will affect the distinction between crops [34,50,51]. How to distinguish between rice, soybean, and corn is the key to paddy rice mapping. The change of red-edge bands is different in the growing stage of crops and how to use the change of the red-edge band to distinguish crops is very meaningful.

The reflectance changes of the two red edge bands are different during the growth period of rice, corn, and soybean, especially the reflectance of the red-edge1 to red-edge2 varies greatly. Therefore, the red-edge integral can be constructed according to the change of the first red-edge band to the near-infrared band in different growth stages of rice, which can effectively distinguish rice, soybean, and corn. REDT method based on red-edge spectral integration and red-edge vegetation index integration can make full use of two red-edge bands for rice mapping.

To verify the advantages of the REDT method, the NNE method was served as a comparison. At the same time, the rice mapping result of GF-6 and GF-1 were compared. The results showed that the accuracy of the REDT method based on GF-6 was significantly higher than that of the NNE method. As shown in Figure 9, when the rice field and soybean field are adjacent, the REDT method can distinguish them well. While the NNE method can easily identify some soybeans as rice, and corn and rice are also easy to be confused (Figure 9). The possible reason is that the reflectance of rice, soybean, and corn varies little from the red band to the near-infrared band during the same period, so it is easy to confuse the three crops by using the spectral integration of these two bands. NDVI can well reflect the growth state of crops when the crops grow from seedlings to just cover the ground surface. However, when the crops cover the surface completely, the NDVI value reaches saturation. Until the crops becoming mature, NDVI cannot reflect the growth state of the crop well [34]. However, the red-edge can distinguish soybean, corn, and rice well in this period. Besides, the REDT method has a good effect on the large-area study site (Jingzhou City) with less noise, which shows that the red-edge characteristic band of GF-6 can effectively carry out large-area rice mapping (Table 3). By comparing the results of GF-6 and GF-1, the red-edge characteristic of GF-6 is more suitable for precision agriculture research. Therefore, the time series-based REDT method can be effectively applied to the mapping of fragmented rice fields in southern China.

*5.2. Application of Red-Edge Band in Rice Growth Monitoring*

At the pre-heading stage, the fitting effect between red-edge VIs (NDRE, NREDI, MTCI, EVI2, $CI_{red-edge}$) and AGB is good, and the highest $R^2$ was 0.90. The change rate of reflectance from the red-edge band to the near-infrared band is significantly higher than that from the red band to the near-infrared band (Figure 7a). Therefore, we consider whether the two red-edge bands of GF-6 could be used to monitor the growth of rice. The fitting effect of the red-edge vegetation index and AGB is better than that of the non-red-edge vegetation index, which indicates that the red-edge band plays an important role in rice growth monitoring. In this paper, our objection is to use GF-6 images for large-area rice growth monitoring. The spatial resolution of the GF-6 image is 16 m, so it is time-consuming and labor-consuming to measure AGB and chlorophyll in the field. How to explore a method to simulate the corresponding AGB of rice is very meaningful. The best model obtained from UAV growth monitoring results was applied to the GF-6 image, and the corresponding AGB of rice on the GF-6 image was obtained. In the study of rice growth monitoring, vegetation index constructed from visible light band to near-infrared band was mostly used [49–51]. Based on the GF-6 image, the red-edge vegetation index was used to monitor rice growth.

At the post heading stage, rice covered the ground completely, chlorophyll accumulated to the maximum, and AGB increased rapidly. However, from the heading stage to the maturity stage, the leaves turned yellow, the chlorophyll content decreased, the vegetation index began to decrease, but the AGB increased continuously. Only using a single vegetation index model cannot monitor the growth of rice well (Figure 14). The partial least squares algorithm combined with multiple VIs cannot only concentrate the contribution of selected vegetation index or red-edge parameters to AGB but also effectively eliminate the collinearity between vegetation index or red-edge parameters. It proves that the PLSR method of multi vegetation index was better than a single vegetation index model in estimating AGB.

According to the monitoring results of rice growth at the pre-heading stage (or post-heading stage), it demonstrated that for different VIs, the improvement of estimation accuracy was different in the two red-edge bands of GF-6 images (Figures 13 and 16). The vegetation index combined with the near-infrared band and the red-edge band1 (CI, NDRE, EVI) has a good effect on the growth monitoring because the difference between the reflectance of the red-edge band1 and the near-infrared band is large. The reflectance difference between the red-edge band2 and the green band is large, so the vegetation index constructed by these two bands (TCI, TCARI) can better reflect the growth of rice, but the $R^2$ of the vegetation index model constructed by the two bands is not as good as the vegetation index model constructed by the red-edge band1 and near-infrared band (Figures 13 and 16). The vegetation index calculated by the two red-edge bands (NREDI) is effective in monitoring rice growth. The results show that the two red-edge bands of the GF-6 image can be well applied to rice growth monitoring research, which provides a new research method and theoretical basis for large area rice mapping and growth monitoring.

The estimated AGB at the pre-heading stage and post-heading stage is divided into four grades, respectively (Figure 17). It can be used to monitor the growth of rice at different levels, and implement management measures for specific fields according to the results of growth grading. Moreover, the rice growth monitoring map made in this paper has a good intuitive and practical value for agricultural technical personnel to guide field production and management. It can be concluded that the rice growth monitoring results generated by GF-6 images can accurately and objectively reflect the rice growth situation.

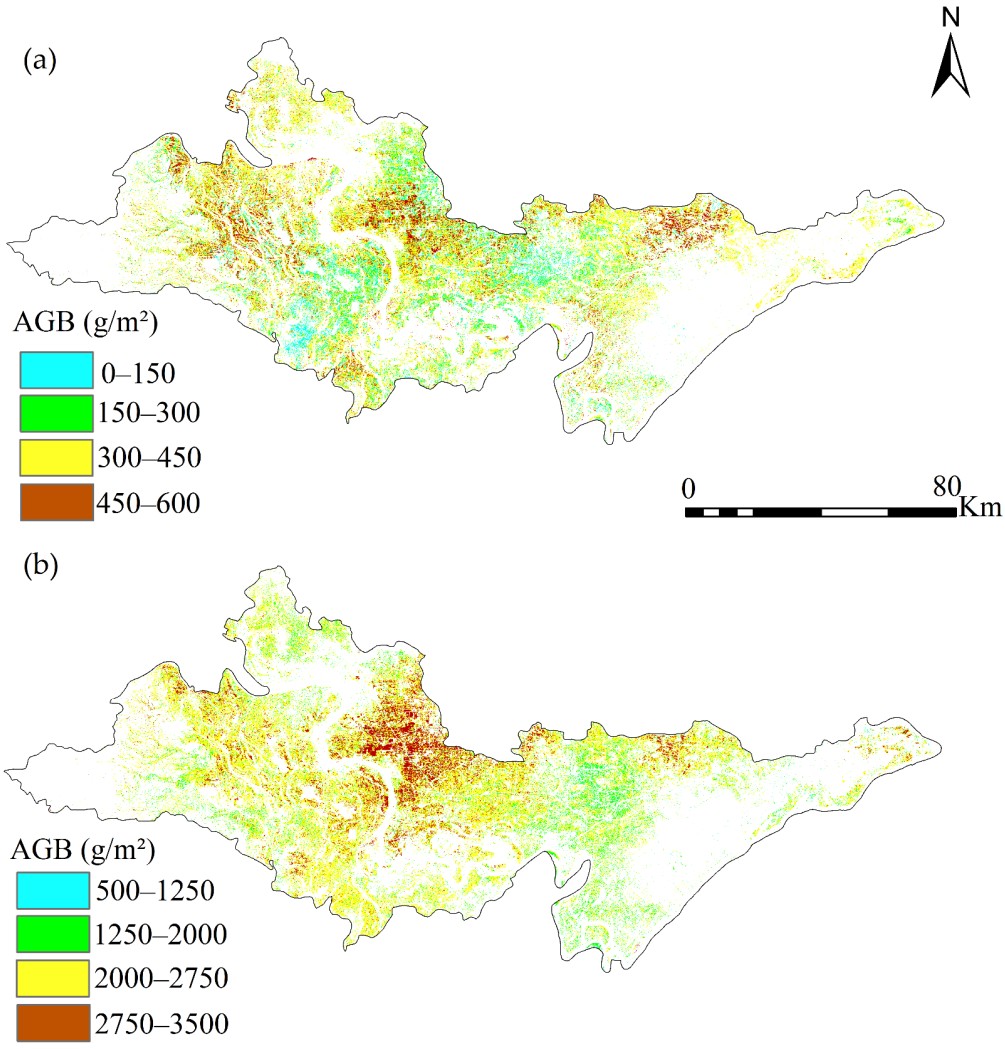

**Figure 17.** Monitoring results of rice growth in pre-heading stage (**a**) and post-heading stage (**b**).

In this paper, the superiority of the GF-6 red-edge band has been brought into full play in rice mapping and growth monitoring. It provides certain technical support and a theoretical basis for the GF-6 image in crop identification and growth monitoring. As different cultivation areas and cultivation measures may cause changes in spectral information, the adaptability of the results to other regions and experimental backgrounds needs to be further verified. In the follow-up precision agriculture research of the GF-6 satellite, we will try more VIs related to the red-edge band, and fully explore the role of the GF-6 red-edge characteristic band in crop accurate recognition.

## 6. Conclusions

In this study, based on GF-6 images, we presented a REDT method based on the combination of time series and red-edge characteristic band. Compared with the traditional phenology and time series method, the REDT method can greatly reduce the confusion effect of crops, and effectively improve the accuracy of rice mapping. It fully shows the importance of the two red-edge bands of GF-6 and their advantages in large area rice mapping. The method is expected to help researchers and agricultural managers of rice mapping in fragmented landscape regions, and it can also provide useful information for agricultural planners (e.g., crop yield estimates, land rational uses). Based on the characteristics of the GF-6 wide range camera, we used two red-edge bands of GF-6 images to monitor rice growth in a large area. The results also proved that the two red-edge bands of the GF-6 image played different roles in rice growth monitoring, and the combination of the two red-edge bands could effectively monitor rice growth.

As GF-6 images provide unprecedented opportunities and challenges in rice mapping, we believe that the strategy based on time series and the red-edge band will be helpful to large area rice mapping and growth monitoring. In future research, we will consider more VIs related to red-edge bands, and fully explore the role of GF-6 red-edge characteristic band in crop accurate identification and growth monitoring, which is of great significance for further research on crop regional mapping and precision intelligent agriculture.

**Author Contributions:** X.J. conceived and designed all the experiments, wrote the main programs and the original draft; S.F. supervised the research, supplemented the experimental algorithm, and provided some meaningful suggestions; X.H. arranged the experimental data and preprocessed the data; Y.L. and L.G. supervised the work and provided financial support for the project. All authors have read and agreed to the published version of the manuscript.

**Funding:** This research was funded by the Piesat Information Technology Co., Ltd. (Beijing, China,2018 project), the National 863 Project of China (2013AA102401).

**Institutional Review Board Statement:** Not applicable.

**Informed Consent Statement:** Not applicable.

**Data Availability Statement:** The data presented in this study are available on request from the corresponding author.

**Conflicts of Interest:** The authors declare no conflict of interest.

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
