# Peer review of "Rice Mapping and Growth Monitoring Based on Time Series GF-6 Images and Red-Edge Bands"

_remotesensing, doi:10.3390/rs13040579_

Round 1

Reviewer 1 Report

Summary - Using GF-1 and GF-6 red edge band to detect rice fields and map above ground biomass. 

General comments-

There are too many poorly written sentences to point out everyone. For example, the entire first paragraph needs rewording. Also it should be “the environment” not “environment (such as in lines 10 an 44). There are several places where the spacing is off (no spaces or spaces in the wrong place). A bunch of the text seems redundant and can be removed. The writting style needs to be improved throughout. 

The word obviously is used way too many times, and does not actually make it "obvious" why the statement is true. 

Line 10- “sustainable development of environment” needs rewording.

Line 11 – I do not think remote sensing itself is an emerging technology, landsat for example has been around since the 1970s and aerial photography before that

Line 12 – “as the sensitive band of vegetation” needs rewording

Line 37-39 – needs reworking

Line 56 – “. Rice fields is a” should be fields are

Line 66 –A single time image is not a time series

Line 87 – should be datesets

Line 100- should say large areas

Line 100 – remote sensing of crops is not a new idea

Line 123 “ red edge band is the sensitive…” this sentence is repeated several times

Line 203 – include the year

Line 218 – add some spaces between the commas

Line 232 – shouldn’t start a sentence with a number

Line 241- 247

Paragraph needs improvement.

Only three field samples were collected?

What was the size of sample collection? Was it a meter quadrant?

Keep is consistent aboveground or above ground

Figure 4 – typo – “cron” should be corn. Maybe add more to the figure header like what the subscripts mean

Figure 6 – What do the colors indicate. What do the red lines indicate

Line 346- explain how they are different.

Lines 351 – 355 and 382 and 393 the numbers should be equations and written as such.

601- why was it irregular? Should explain why higher above ground biomass values relate to the lower VI

640 – fix GF-6

643 – should be vegetation indices  

652- give values for NEE method

Author Response

Thank you for your suggestions and comments. Please see the attachment。

Reviewer 2 Report

Review of the paper “Rice Mapping and Growth Monitoring with GF-6 Images, Red-edge Based and Time Series” send to Remote Sensing.

The article proposes a methodology to improve the classification of rice fields and their growth using GF-6 satellite images. The paper is interesting and suitable for publication in RS, but several aspects need to be improved.

General comments:

Some aspects are not well explained and details are missing, especially from the field data collection. The use of UAVs is not understood because the same results can be achieved by directly calibrating and validating the field data with the satellite GF-6 spectra. At the same time, there are text and graphics that can be reduced to make the article shorter and highlight the most important results.

Specific comments.

There are missing separating spaces in all the text, for example on line 52 “likelihood[12],support vector machine (SVM)[13,14], artificial neural network(ANN)[15], and” should be “likelihood_[12],_support vector machine (SVM)_[13,_14], artificial neural network_(ANN)_[15], and”. REVIEW ALL TEXT.

Line 24: define NNE

Line 46: Sentinel-2 et al) >> Sentinel-2 et al.)

Line 79 : 290m > 290 km

Line 81: on sentinel-2 MSI image, sentinel-2 NDVI… > on Sentinel-2 MSI image, Sentinel-2 NDVI…

Line 92: define WFV

Line 95: above ground biomass (AGB) > above ground dry biomass (AGB)

Line 104: define BP

Line 118: Lai > LAI

Line 151: remove the reference to Table 1, as Table 1 is about GF-6

Line 163: define WFV (or in line 92)

Line 169: You say that 9 images were used, but it is not clear if they are 9 of the same date to build a mosaic or they are different dates. The numbers do not correspond to those in table 1 (column NUMBER). Please specify better. It would also be good to see in Figure 1 if the GF-6 scenes (tiles) cover the entire area under study, or how the satellite scenes appear.

Line 172: What the NUMBER column refers to?

Line 188 Table 1: The columns SPATIAL RESOLUTION and SWATH WIDTH are not necessary, because it is the same value. You only need to explain it in the text.

Line 190: I propose to delete this section. I believe that the study can be done by directly relating the field data to that of the GF-6 satellite. In addition, one of the two bands of the Red edge on which the whole study is based does not coincide in the UAV and the satellite.

Line 202: If it is included, you have to indicate the pixel size.

Line 235: … “A total of 2197 samples (1377841pixels)”… but in line 232 it says 1300 field points. Explain better how much data has been used. Pixels are from UAV or from satellite?

Line 245: The measurement process has been done on the same dates as the satellite images? Specify when the field data was taken.

Line 268: “(2)On the image of 27 July, the NDVI value is set to be less than 0.45.” What is this condition used for?

Line 285 Figure 5: The size of the marks is very small, you can not read the dates. Specify in the text if these graphics are only from one pixel or an average of several pixels (or a region of interest, ROI). From UAV or satellite?

Line 298 Figure 6: Eliminate the last column (OCTOBER), it does not contribute anything and allows to make the figure (and text) bigger to see it better.

Line 303: Specify better how figure 7 has been obtained. One or several pixels, as they have been selected?

Line 321: formula 6: What is Bi? Are wavelengths? Specify, the same in formula 7.

Line 324: “represents the integral value of time t from the red edge band1 to the near infrared band” this sentence is repeated (lines 323 and 324) and is not correct, explain better the difference between RE_Si and Si.

Line 330, formula 8: must be Vi+Vj?

Line 343 Figure 7: put the spectra in Figure 7a by writing wavelengths (not number band) on the X axis, so it will be easier to interpret them. Now they are in disorder.

Line 349-350: write better, the verb is missing.

Line 351: the terminology is unclear. If it refers to an interval, it is better to put the condition only with a numerical value, for example RE_S0727 > 34   and  RE_S0812 > 38 and remove a8, a9, a10…  

Line 353: August to August was significantly… >> August to 20 August was significantly…

Line 367: define NNE

Line 381: explains better what this threshold is used for. Also explain if ALL conditions must be satisfied.

Line 392: you have to specify which index is used in   S0815 < 25

Line 424, Tables 2 and 3: If the UAV is maintained, unify these two tables into a single one (the name of the indices and the references are the same, only the bands are different).

Line 499: “…the above-ground biomass(AGB)” >> “…the above ground dry biomass(AGB)”

Lines 505-508 : The MTCI index that is used to simulate AGB uses a band (720 nm) that is not on GF-6, so the UAV calibration cannot be used with GF-6. It is not understood why it is not calibrated directly by correlating the AGB field data with the GF-6 spectra.

Line 512-517: Remove figure 13 and 14 and replace with another with indices from GF-6  versus AGB field data.

Lines 519-532 and Figures 15 and 17 : the same for the post heading stage, remove figure 15 and 17 and replace with another with indices from GF-6  versus AGB field data.

Line 547: If the R2 values are clearly shown in the graphs, Table 6 can be removed.

Line 637 figure 18: Add the units of AGB in the maps.

Line 640: for gf-6… >> for GF-6…

Line 646: Conclusion >> Conclusions

Line 676: Reference >> References

Lines 676-831 : Check the bibliography. There are some repeated ones (e.g. 8 and 19 or  60 and 62). In some citations the authors are missing (e.g. 46 or 52). Review all to comply with the rules.

Author Response

(The authors gave the same response as above.)

Reviewer 3 Report

This paper is about a new red-edge decision tree algorithm to map and monitoring rice growth, based on GF-6 satellite time-series.

General comments:

The document is very interesting and well supported by graphs and analysis, but there are a few redundancies and repetitions, and it is not clearly written. There is some confusion in the presentation of methodology and main results obtained, making the paper difficult to read.  
A general check of all manuscript is needed in order to improve its readability.
Some figures have low-quality, please replace them with high-quality figures.

Specific comments:

  • Line24: Please add the full acronym NNE;
  • Lines 62-66: Please move to discussion section;
  • Line 76: the authors list the satellite with red-edge band sensors. Pleas add WorldView-3;
  • Line 191: Please add specifications about used UAV;
  • Line 202: Please add the obtained GSD of the UAV images
  • Lines 204-205: The authors say that a band-to-ban registration has been done before flight. Please add few lines about this process;
  • Figure 2: the authors used a multispectral camera with 12 separate sensors but the one showed in figure has only 6 separated sensors (Tetracam micro-MCA 6). Please replace with the correct camera (Tetracam micro-MCA 12). Moreover, please add the position on the field of the 8 calibration target used to compute radiometric calibration;
  • Lines 275-278: Please use the sensor central band reflectance values instead of “green”, “red” and “nir”;
  • Lines 300-318: please move to results section;
  • Lines 501-503: please add to central band values also the bandwidth;
  • Lines 650-655: please move to results section.

Author Response

Thank you for your suggestions and comments. Please see the attachment.

Round 2

Reviewer 2 Report

The paper has been significantly improved and can be published in the current form.

Author Response

Thank you  for your valuable suggestions and comments.

Reviewer 3 Report

Dear authors,

thankyou for the effort in taking into consideration the comments.

All points have been adequately addressed.

Author Response

(The authors gave the same response as above.)
